# Sleep Parameters and Quality of Life in Children with Monosymptomatic Nocturnal Enuresis: Association, Prediction and Moderation Analysis

**DOI:** 10.3390/ijerph22050765

**Published:** 2025-05-13

**Authors:** Leticia Soster, Renatha Rafihi-Ferreira, Simone Fagundes, Adrienne Lebl, Vera Koch, Ila Linares

**Affiliations:** 1Pediatric Division, Hospital das Clínicas, University of São Paulo, São Paulo 01246-903, Brazil; leticia.soster@hc.fm.usp.br (L.S.); sifagun@terra.com.br (S.F.); adrienne.lebl@gmail.com (A.L.); vera.koch23@gmail.com (V.K.); 2Department of Clinical Psychology, University of São Paulo, São Paulo 05508-030, Brazil; ilamplinares@gmail.com

**Keywords:** monosymptomatic nocturnal enuresis, behavioral problems, polysomnography, quality of life

## Abstract

Although often considered benign, monosymptomatic nocturnal enuresis (MNE) can significantly affect various domains of children’s quality of life. This study aimed to investigate the relationship between polysomnography-derived sleep parameters and quality of life in children with MNE. Forty-four participants aged 8–12 years, with a diagnosis of MNE, underwent polysomnography and completed a quality of life questionnaire (PedsQL). Multiple linear regressions were used to assess the role of sleep variables in predicting four PedsQL domains. Moderation analysis was also performed to evaluate whether the N2 sleep percentage influenced the relationship between arousals and PedsQL. A higher frequency of microarousals predicted lower scores in the physical (B = −3.28, *p* = 0.01) and social (B = −3.50, *p* = 0.004) domains. A higher N2 sleep percentage was associated with better social (B = 0.69, *p* = 0.03) and school functioning (B = 0.88, *p* = 0.03). Moderation analysis revealed that N2 sleep amplified the negative impact of arousals on the social domain, particularly when the N2 sleep percentage was one standard deviation above the mean (B = −4.31, *p* < 0.001). The findings underscore the importance of sleep continuity and N2 stability for the quality of life of children with MNE. These results demonstrate the complex interaction between sleep and quality of life in pediatric enuresis.

## 1. Introduction

Monosymptomatic nocturnal enuresis (MNE) is a common childhood disorder that is characterized by recurrent episodes of nocturnal urinary incontinence in children without signs of daytime voiding dysfunction or other urological comorbidities [1]. Although often considered a benign and self-limiting condition, MNE can have significant impacts on the child’s physical, emotional, and social well-being, especially when it persists beyond the expected age of toilet training.

There is evidence suggesting alterations in the sleep patterns of children with MNE, including fragmentation, decreased sleep efficiency, and changes in sleep architecture, particularly during slow-wave and REM sleep [2,3]. Sleep deprivation and poor sleep quality, in turn, are correlated with cognitive and emotional deficits, which can compromise the academic performance, emotional regulation, and social interaction of these children [4]. Furthermore, the impact of MNE extends beyond nighttime difficulties, negatively affecting self-esteem, increasing anxiety, and limiting participation in social activities such as sleeping away from home [5]. These factors indicate how much the challenges faced by these children affect different aspects of daily life; it is therefore fundamental to assess the impact of MNE from a broader perspective, including its effects on quality of life.

Quality of life is a multidimensional and subjective concept that has been widely discussed in the scientific literature; however, there is no universally accepted definition. Although no consensus exists in the literature on the term quality of life, the World Health Organization (1995) [6] defines it as individuals’ perception of their position in life in the context of the culture and value system in which they live and in relation to their goals, expectations, standards, and concerns. In the case of children, this perception can be assessed by both the child and their caregivers considering different domains, including the child’s physical, psychological (cognitive and emotional), and social domains. Within this context, studies have shown that children with MNE have a lower quality of life than non-enuretic children, reinforcing the need for early and integrated therapeutic approaches to minimize the adverse effects of this condition [7,8].

Although authors have investigated the impact of MNE on quality of life, we found no studies that directly correlated polysomnography (PSG) parameters with quality of life measures. The available studies were based on indirect sleep measures such as actigraphy rather than PSG, a fact that may limit their accuracy in identifying more subtle changes in sleep architecture. This gap in the literature highlights the need to explore how objective changes in sleep macrostructure are associated with the subjective perceptions of physical, emotional, and social well-being of these children. Understanding the relationship between sleep structure and quality of life in these patients can contribute to the improvement of therapeutic strategies in biopsychosocial terms since this knowledge considers the neurophysiological aspects of sleep and the psychosocial impact of the condition.

Given this scenario, the aim of the present study was to investigate the relationship between polysomnographic variables and quality of life in children with MNE. We sought to analyze whether sleep parameters can be used to predict scores across the different domains of the Pediatric Quality of Life Inventory (PedsQL) [9]. The study also evaluated the moderating role of N2 sleep percentage in the association between microarousals and the social domain of quality of life. Our hypothesis is that changes in sleep architecture influence different PedsQL domains, particularly those related to academic performance and emotional well-being.

## 2. Methods

This was a cross-sectional study whose data were derived from a randomized clinical trial that evaluated the efficacy of three treatments in MNE patients [10]. Details of the study procedures have been published previously [10]. The randomized clinical trial was conducted by a multidisciplinary team of the Pediatric Nephrology and Sleep Physiology Unit of the Children’s Institute, University Hospital, University of São Paulo Medical School (HC-FMUSP), Brazil. The study was approved by the local Ethics Committee (Approval Number: 0649/10) and informed consent was obtained from all participants.

The families were invited to participate in the randomized clinical trial through lay press releases. Eligible participants underwent pre- and post-treatment full-night video-monitored PSG [11]. In addition, the participants underwent multidisciplinary clinical evaluation and quality of life-related complaints were assessed using the PedsQL scale.

### 2.1. Participants

Criteria for inclusion were an age between 6 and 17 years; a diagnosis of MNE according to the criteria of the International Children’s Continence Society [1], including at least two episodes of nocturnal enuresis per month; full daytime urinary continence; absence of renal, urological or neurological diseases; no specific prior treatment (pharmacological or behavioral) of enuresis episodes; completion of the study protocol.

Children with a clinical diagnosis of any disease or condition that could affect sleep; children with renal, urological, neurological, or psychiatric disorders; children with daytime lower urinary tract symptoms (incontinence or urgency) were excluded [10].

### 2.2. Polysomnographic Recording

The PSG examinations were performed at the Pediatric Sleep Laboratory of the Children’s Institute, HC-FMUSP, in a quiet room under video monitoring accompanied in a separate room by a specialized technician.

All recordings started at the habitual bedtime of the patients and continued until spontaneous awakening. Electroencephalogram (EEG) recordings and electrode placement were performed according to the 10–20 system. The PSG assembly included six EEG channels [Fp1-A2, Fp2-A1, C3-A2, C4-A1, O1-A2, and O2-A1], left and right electrooculography, a chin electromyogram, an electrocardiogram, left and right tibialis anterior muscle electromyogram, a nasal cannula, and measurement of thoracic and abdominal respiratory effort and oxygen saturation.

A board-certified clinical neurophysiologist visually analyzed all recordings. Standard sleep parameters were derived and tabulated for statistical analysis. Sleep data were stored on a hard disk using a digital PSG system (Embla N7000, Medcare, Iceland) and the corresponding software (Remlogic, Medcare, Iceland).

### 2.3. Sleep Macrostructure

Sleep was subdivided into 30 s epochs and the sleep stages were classified according to the criteria of the American Academy of Sleep Medicine. Arousals were identified as two or more consecutive epochs classified as wakefulness, surrounded by epochs of sleep [12].

The following conventional sleep parameters were assessed: sleep efficiency defined as the percentage of total sleep time in relation to total time in bed; wakefulness after sleep onset (WASO) defined as the percentage of time spent awake after sleep onset; sleep latency defined as the time from lights out to sleep onset; total sleep time (TST) defined as the time between sleep onset and the end of the last sleep epoch, excluding awake time. Additional sleep parameters included the percent distribution of TST in sleep stages N1 (%), N2 (%), N3 (%), and REM sleep (%), as well as the limb movement index (number of limb movements divided by TST), periodic limb movement index (number of periodic limb movements divided by TST), and arousals (number of awakenings divided by TST).

It is important to clarify that, although we initially refer to sleep fragmentation using a broader conceptual framework that includes brief microarousals to emphasize the clinical relevance of disrupted sleep, the scoring of arousals in this study was performed strictly according to the American Academy of Sleep Medicine (AASM) Scoring Manual (version 3.0) criteria. This methodological rigor ensures that the associations reported between sleep parameters and quality of life are based on standardized and validated measures.

### 2.4. Instruments

The PedsQL 4.0 was developed by Gayral-Taminh et al. (2005) [9] and validated for Brazil by Assumpção et al. (2000) [13]. The instrument consists of 23 items distributed across four domains: social, school functioning, physical, and emotional. All domains are scored on a scale from 0 to 100, where higher scores indicate better quality of life.

### 2.5. Data Analysis

Descriptive analysis of the variables was performed, calculating means, standard deviations, and proportions. A level of significance of 0.05 was adopted for all comparisons. The normality of the distributions was assessed using the Shapiro–Wilk test.

Pearson’s correlation coefficient between continuous variables was applied to examine the association between polysomnographic parameters and quality of life domains. For cases with missing data, listwise deletion was used so that only participants with complete information were included in the subsequent analyses.

Multiple linear regression models were used to assess the predictive effect of the sleep variables—including N2 and N3 percentage, number of microarousals, movement index per hour, and WASO—on the four PedsQL domains (physical, emotional, social, and school functioning). The models consisted of combinations of the N2 or N3 variables with the other predictors, totaling eight independent models, two per PedsQL domain, each containing N2 or N3 (Appendix A).

The assumptions of the models were assessed using analysis of residuals, homoscedasticity, multicollinearity (variance inflation index < 5), absence of influential outliers (Cook’s distance < 1), and independence of residuals (Durbin–Watson test).

Additionally, moderation analysis was performed to determine whether the N2 variable modified the relationship between microarousals and the social domain of the PedsQL scale. The significance of the interaction effect was estimated by linear regression with cross-over interaction. Simple effects were evaluated based on standard deviations of the mean of the moderating variable. The N2 stage was chosen as a moderator variable based on previous studies such as that by Soster et al. [14]. These authors demonstrated reorganization of sleep architecture after effective treatment of enuresis, with a reduction in N2 sleep and an increase in N3 sleep, suggesting that N2 acts as an intermediate and adaptive stage. Thus, the inclusion of this variable as a moderator is aimed at understanding whether the percentage of time spent in this stage influences the relationship between sleep fragmentation and the social domain of quality of life perceived by children with MNE.

## 3. Results

### 3.1. Baseline Clinical and Sociodemographic Characteristics of the Participants

The PSG and PedsQL results of 44 patients (12 female) were analyzed. Of these, all had MNE (39 primary and 5 secondary) and underwent all steps of the study. Enuretic episodes were observed during all the PSG recordings. Table 1 shows the demographic and polysomnographic characteristics of the participants.

### 3.2. Correlations Between Sleep Parameters and PedsQL Subscales

Pearson’s correlation analysis revealed statistically significant associations between PedsQL scores and the polysomnographic parameters. Higher scores in the social domain were moderately associated with a lower frequency of microarousals during the night (*r* = −0.373, *p* < 0.05) and with a higher N2 sleep percentage (*r* = 0.300, *p* < 0.05). Furthermore, higher scores in the physical domain were also associated with a reduction in the frequency of microarousals (*r* = −0.353, *p* < 0.05), indicating that a better perception of physical quality of life is related to greater sleep continuity (Table 2).

### 3.3. Predictors of Quality of Life

The results showed that microarousals were significant predictors of the physical domain in the models that included either N2 or N3 sleep as the independent variable. Specifically, we observed that each additional occurrence of microarousals was associated with an average reduction of 3.28 points in the physical domain score (B = −3.28, *p* = 0.01).

In the social domain, each 1% increase in N2 sleep percentage increased the social score by 0.69 points (B = 0.69, *p* = 0.03), while each additional occurrence of microarousals reduced the social subscale score by 3 points (B = −3.5, *p* = 0.0004). In the model including N3, each occurrence of microarousals reduced the social subscale score by 3 points (B = −3.34, *p* = 0.008).

In the school functioning domain, each 1% increase in N2 sleep increased on average the score of this subscale by 0.88 points (B = 0.88, *p* = 0.03). No statistically significant findings were obtained for the models including the school functioning domain and N3, or the emotional domain and N2 or N3.

### 3.4. Moderation Analysis of Quality of Life

Finally, moderation analysis was performed, which indicated that the N2 sleep percentage exerts a moderating effect on the relationship between the frequency of microarousals and social domain scores of the PedsQL scale. The interaction effect was statistically significant (B = −0.40, *p* = 0.007), suggesting that the negative impact of microarousals on social quality of life varies as a function of the N2 sleep percentage.

The analysis of simple effects revealed that, when the N2 sleep percentage is one standard deviation above the mean, the increase in the number of microarousals is associated with a marked reduction in social domain scores (B = −4.31, *p* < 0.001), indicating an amplified effect of sleep fragmentation in individuals with a higher N2 sleep percentage (Figure 1).

## 4. Discussion

The findings of the present study contribute to the understanding of the effects of sleep architecture on the quality of life of children with MNE. The analysis of polysomnographic parameters showed that variables such as microarousals and the time spent in N2 sleep are associated with specific quality of life domains measured by the PedsQL scale, particularly the social domain. Furthermore, microarousals were predictors of the physical and social domains, and N2 sleep had a positive effect on social and school functioning scores. Moderation analysis revealed that N2 sleep percentage intensifies the negative impact of microarousals on social functioning, demonstrating a complex interaction between sleep continuity and sleep composition in the subjective experience of quality of life in children with MNE.

The negative correlation observed between the microarousal index and physical and social scores reinforces the hypothesis that sleep fragmentation has a deleterious effect on multiple spheres of daytime functioning. Frequent microarousals compromise sleep continuity and reduce the restorative efficiency of sleep, which can increase fatigue, reduce physical fitness, and lead to difficulties in social interactions [15]. These findings are consistent with studies that have associated sleep fragmentation with poorer mood regulation, increased stress reactivity, and impaired social responsiveness in pediatric populations [15,16]. The existence of this association even in children without evident neurological disorders suggests the microarousal index to be a sensitive marker of functional risk, with clinical implications for the assessment and management of sleep in contexts of typical development.

The present results highlight the functional role of N2 sleep in the quality of life of children and adolescents with MNE, especially in the social domain. Although classified as light sleep, previous studies have already suggested the participation of this stage in sensory consolidation and memory integration processes, which may favor social performance in children [17,18]. The positive association between a higher N2 sleep percentage and better indicators in these domains suggests that preserving this stage may protect against the effects of sleep fragmentation. Regarding N3 sleep, the present data do not support a direct relationship between its percentage and the PedsQL findings.

Alterations in N2 sleep architecture have also been reported in other parasomnias, such as disorders of arousal, including sleepwalking and confusional arousals. Studies have suggested that increased sleep instability, reflected in microarousals and cyclic alternating patterns during N2 sleep, may predispose individuals to episodes of behavioral dysregulation during sleep. Although our study did not directly assess CAP parameters, the observed changes in N2 sleep among children with MNE may reflect a broader pattern of sleep fragmentation and instability, partially shared with other parasomnias [19]. This possible overlap warrants further investigation in future studies.

Recent studies reinforce the role of sleep fragmentation in the daytime impairments observed in children with MNE. Sekerci et al. (2025) reported that increased microarousals were associated with cognitive and psychosocial deficits [20], while Adisu et al. (2025) found that sleep instability correlated with poorer school and social functioning, independent of enuresis severity [21]. These findings support the hypothesis that microarousals during lighter sleep stages, particularly N2, contribute to broader functional impairments, consistent with the present results.

The close to significant correlation between the school functioning subscale and WASO raises an interesting possibility, i.e., increased wakefulness after sleep onset may be related to cognitive or behavioral adjustments that affect school performance. However, this trend should be interpreted with caution, given the borderline *p* value and the lack of correlations with other sleep measures.

Regarding the analysis of predictors, although some studies including those involving the present cohort did not identify significant differences in sleep macrostructure between children with enuresis and controls [11], longitudinal pre- and post-treatment assessment revealed changes suggestive of a functional role of N3 sleep in patients showing clinical improvement [14]. Specifically, an increase in the percentage of N3 sleep and a concomitant reduction in N2 sleep were observed, indicating a possible adaptive reorganization of sleep architecture after the intervention. These findings reinforce the hypothesis that the N2 stage may act as an intermediary in the transition to slow-wave sleep (N3), exerting a regulatory function in the deepening of sleep [22,23].

In the present study, the functional role of N2 sleep was explored by moderation analysis, which indicated a significant effect on the relationship between the frequency of microarousals and the social domain of the PedsQL scale. Specifically, we found that the negative impact of microarousals on this quality of life domain was more pronounced in individuals with a higher N2 sleep percentage, suggesting that this stage may represent a critical moment of vulnerability to sleep fragmentation. Although traditionally classified as light sleep, N2 sleep appears to play important roles in sensory consolidation and in the regulation of interpersonal processes. This fact may explain the association of this stage with adverse social outcomes in contexts of interrupted sleep [17,18].

Although N3 sleep has been widely recognized for its restorative role, the present results indicate that its percentage did not exert a significant effect on any of the domains of children’s quality of life analyzed. Even in the models in which microarousal were found to be important negative predictors of physical and social scores, N3 sleep did not contribute significantly to the explained variance of these outcomes. This fact suggests that the mere presence of deep sleep is not sufficient to compensate for the deleterious effects of sleep fragmentation. According to the literature, N3 sleep is involved in essential functions such as metabolic and immunological recovery and neuroendocrine regulation [24,25], as well as in processes of synaptic plasticity during development [26]. However, the present data reinforce the hypothesis that the functional efficacy of N3 sleep is compromised by repeated interruption of the sleep–wake cycle, with sleep continuity being a critical factor for the full manifestation of its benefits.

With respect to quality of life, the losses observed in the social domain must be interpreted in light of the psychosocial challenges imposed by enuresis, such as the fear of sleeping away from home, stigmatization by peers, and family embarrassment [27]. Although we found no significant polysomnographic predictors for the emotional domain of the PedsQL scale, it is possible that emotional aspects act as mediators in the relationship between sleep and quality of life, influencing, for example, how the child interprets or responds to social difficulties. Therefore, the assessment of quality of life in children with MNE should consider not only clinical symptoms but also daily functioning in multiple contexts—school, family, emotional, and social. Finally, although treatment usually focuses on resolving urinary symptoms, the data suggest that strategies designed to promote sleep quality and continuity may have direct positive effects on the child’s physical and interpersonal functioning.

This study has some limitations that must be considered when interpreting the results. Although clinically and carefully characterized, the sample size was relatively small. No formal power analysis was performed prior to participant recruitment; thus, the sample size was determined based on feasibility considerations. This fact can limit generalization of the findings and reduce the statistical power to detect smaller effects, especially in interaction analyses. In addition, the cross-sectional design of most analyses limits causal inferences, particularly regarding the direction of the associations between sleep fragmentation and quality of life.

On the other hand, this study has important methodological strengths. The selection of patients from a population without a history of diseases, the careful and multidisciplinary assessment, and the exclusion of any other psychiatric, sleep, nephrological, or neurological disorders ensure an adequate sample for functional analyses. The polysomnographic analysis was conducted with technical rigor, permitting the accurate extraction of micro- and macrostructural variables. The inclusion of moderation analysis is an advancement in the understanding of the interactions between sleep stages and fragmentation, providing a more refined functional perspective. Furthermore, the use of a validated and multidimensional instrument for quality of life assessment allowed us to capture relevant nuances of children’s physical, social, school, and emotional functioning, aligning the outcomes of the study with clinically significant measures.

Taken together, these findings reinforce the importance of considering not only the amount and stages of sleep but also sleep continuity and stability as key elements in promoting child welfare.

## 5. Conclusions

The present study demonstrated that sleep fragmentation, measured by the frequency of microarousals, is negatively associated with physical and social quality of life in children with MNE, regardless of the percentage of N3 sleep. Furthermore, the N2 stage was found to moderate the deleterious effects of fragmentation, suggesting that, although essential for sensory consolidation and integration processes, this stage may represent a window of vulnerability to sleep interruption. The present results highlight the need for therapeutic approaches aimed not only at expanding the restorative stages of sleep but also at their continuous preservation in order to promote overall health and daytime functioning.

## Figures and Tables

**Figure 1 ijerph-22-00765-f001:**
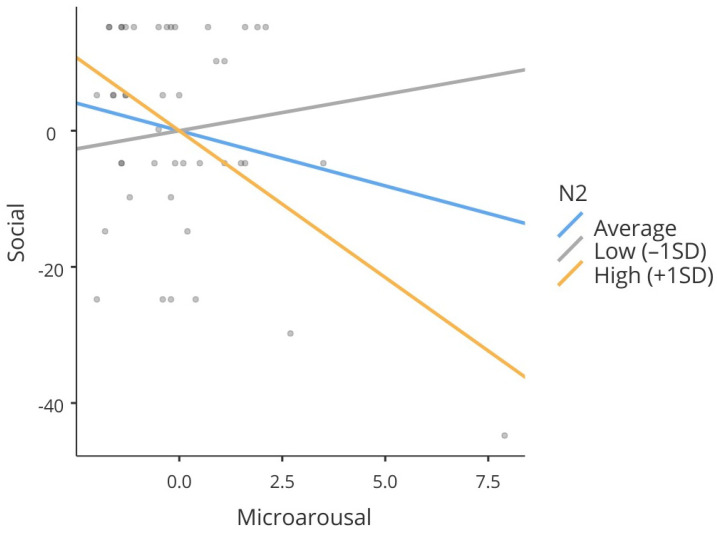
Moderation analysis.

**Table 1 ijerph-22-00765-t001:** Demographic and polysomnographic characteristics of the participants.

	Pretreatment (n = 44) Median [IQR]
Sex	12 F:32 M
**Polysomnography data**	
Apnea–hypopnea index	0.85 [0.375–1.4]
Age	10 [8–12]
Sleep efficiency	92.55 [86.925–95.9]
Sleep latency	13.95 [5.5–27.2]
N1 percentage	3.8 [2.2–5.275]
N2 percentage	57.35 [52.45–61.775]
N3 percentage	23.7 [20.05–28.375]
REM sleep percentage	14.25 [9.5–18.575]
WASO	32.55 [17.625–59.775]
Movement index	7 [4.9–9.9]
Periodic limb movement index	0.25 [0–0.6]
Arousals	1.75 [0.675–2.75]
SpO_2_	95.65 [94.575–96.3]
**PedsQL subscales**	
Physical score	78.125 [68.75–89.063]
Emotional score	67.5 [50–90]
Social score	90 [80–100]
School functioning score	75 [60–85]

**Table 2 ijerph-22-00765-t002:** Pearson correlation between PedsQL data and polysomnographic variables.

	Arousal Index	N2%	N3%	WASO	Movements/hour
Physical score	−0.353 *	0.222	−0.252	0.049	0.020
Emotional score	−0.240	−0.057	0.002	−0.100	−0.057
Social score	−0.373 *	0.300 *	−0.203	−0.114	0.084
School functioning score	−0.146	0.295	−0.226	0.121	−0.085

Note. * *p* < 0.05.

## Data Availability

The original contributions presented in this study are included in the article. Further inquiries can be directed to the corresponding author.

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
