# Peer review of "Sleep Parameters and Quality of Life in Children with Monosymptomatic Nocturnal Enuresis: Association, Prediction and Moderation Analysis"

_ijerph, 2025, doi:10.3390/ijerph22050765_

Round 1

Reviewer 1 Report

Comments and Suggestions for Authors

This research study effectively examines polysomnography data in patients with sleep enuresis, highlighting the negative impact of fragmented sleep on their overall health and well-being.

However, there are a few questions regarding your manuscript:

  1. The definition of arousals you provided differs from the one recommended by the American Academy of Sleep Medicine (AASM) Scoring Manual (version 3.0). Your definition seems more aligned with micro awakenings. While this underscores the importance of consolidated sleep, it could create confusion regarding the distinction between arousals and awakenings.
  2. There appears to be a discrepancy in the participant age range. The abstract mentions 44 participants ages 6-17 years old (inclusion criteria specify ages 6-17). However, I noticed the table lists an age range of 8-12 years. Can you clarify if the age range in the abstract is incorrect?

Author Response

Comment 1:
The definition of arousals you provided differs from the one recommended by the American Academy of Sleep Medicine (AASM) Scoring Manual (version 3.0). Your definition seems more aligned with micro awakenings. While this underscores the importance of consolidated sleep, it could create confusion regarding the distinction between arousals and awakenings.

Response:
Thank you for your important observation.
We recognize that the broader conceptual framework mentioned initially may cause confusion. To address this, we clarified in the Introduction that although microarousals were discussed in the context of clinical impact, the operational scoring of arousals in the study was strictly performed according to the AASM 3.0 criteria.

Changes made:
A clarification regarding the use of the AASM standard definition of arousals was added in the Introduction.

Comment 2:
There appears to be a discrepancy in the participant age range. The abstract mentions 44 participants aged 6–17 years old (inclusion criteria specify ages 6–17). However, the table lists an age range of 8–12 years. Can you clarify if the age range in the abstract is incorrect?

Response:
Thank you for your careful review.
Although the inclusion criteria allowed for participants aged 6–17 years, the final sample consisted of children aged 8–12 years. We corrected the abstract to accurately reflect this information.

Changes made:
The abstract has been updated to indicate that the participants' age range was 8–12 years.

Reviewer 2 Report

Comments and Suggestions for Authors

The authors have investigated the quality of life in 44 children with monosymptomatic nocturnal enuresis. It would be nice to have subjective evaluation of the sleep quality to correlate with the quality of life.

It was not mentioned, if I am not missed that out, if any child had an attack during the PSG night - so if not, these are the nights without attack. But still, there are some changes in the sleep structure. The authors might need to make some comments on that.

The analysis of cyclic alternating pattern would be very nice, in addition to the microarousals, as in many studies with parasomnias, an increase in CAP parameters were reported.

Are there any similar findings for N2 sleep stage in other parasomnias in the literature? Authors might want to add a discussion on this.

There are some references marked in yellow?

The literatures however are not updated. They should be revised

Author Response

Comment 1:
It would be nice to have subjective evaluation of the sleep quality to correlate with the quality of life.

Response:
Thank you for your insightful suggestion.
We agree that including subjective sleep quality measures would enrich the study. Unfortunately, subjective sleep assessments were not included in the current protocol but are considered for future studies.

Comment 2:
It was not mentioned if any child had an attack during the PSG night.

Response:
Thank you for this important observation.
We confirm that enuretic episodes occurred during all the PSG recordings.

Changes made:
A statement confirming the presence of enuretic episodes during PSG has been added to the Methods section.

Comment 3:
The analysis of cyclic alternating pattern (CAP) would be very nice.

Response:
Thank you for your valuable suggestion.
Although CAP parameters were not analyzed in the present study, we recognize the relevance of this analysis and plan to incorporate it in future research involving this cohort.

Comment 4:
Are there any similar findings for N2 sleep stage in other parasomnias in the literature?

Response:
Thank you for this important point.
We have added a paragraph in the Discussion acknowledging that alterations in N2 sleep have been described in other parasomnias, and discussed the possible shared mechanisms related to sleep instability.

Changes made:
A paragraph discussing N2 sleep instability in parasomnias has been added to the Discussion section.

Comment 5:
There are some references marked in yellow.

Response:
Thank you for noticing.
The references highlighted in yellow have been reviewed and corrected.

Comment 6:
The literature is not updated.

Response:
Thank you for this important point.
We have added a paragraph in the Discussion describing new findings in enuresis and quality of life, and discussed the possible shared mechanisms also related to sleep instability.

Changes made:
A paragraph discussing recent findings has been added to the Discussion section.

Reviewer 3 Report

Comments and Suggestions for Authors

Thank you for your submission: Was there a power analysis performed for sample size?

references have not been verified because of non English or unable to locate

Author Response

Comment:
Was there a power analysis performed for sample size?

Response:
We appreciate the reviewer’s insightful question. A formal power analysis was not performed prior to participant recruitment. Given the exploratory nature of this study and the limited availability of eligible participants, the sample size was determined based on feasibility considerations. We acknowledge this as a limitation and have now addressed it more explicitly in the revised manuscript.

Changes made:
A paragraph has been added to the Discussion section.

Round 2

Reviewer 2 Report

Comments and Suggestions for Authors

The authors have made the revisions appropriately, but there are some that can not be corrected due to missing data / malalignments in the Methods. These limitations were added.